# Patient and Staff Perceptions on Using Bioelectrical Impedance Analysis in an Outpatient Haemodialysis Setting: A Qualitative Descriptive Study

**DOI:** 10.3390/healthcare10071205

**Published:** 2022-06-27

**Authors:** Kristin Gomes, Ben Desbrow, Chris Irwin, Shelley Roberts

**Affiliations:** 1School of Health Sciences and Social Work, Griffith University, Gold Coast Campus, Southport, QLD 4222, Australia; kristin.gomes@griffithuni.edu.au (K.G.); b.desbrow@griffith.edu.au (B.D.); c.irwin@griffith.edu.au (C.I.); 2Menzies Health Institute Queensland, Griffith University, Gold Coast Campus, Southport, QLD 4222, Australia; 3Allied Health Research, Gold Coast Hospital and Health Service, 1 Hospital Blvd, Southport, QLD 4219, Australia

**Keywords:** acceptability, bioelectrical impedance, body composition, malnutrition, haemodialysis

## Abstract

Bioelectrical impedance analysis (BIA) is an objective hydration and body composition assessment method recommended for use in haemodialysis patients. Limited research exists on the acceptability and utility of BIA in clinical practice. This qualitative study explored patient and staff acceptability and perceived value of BIA in an outpatient haemodialysis setting at a tertiary public hospital in Queensland, Australia. Participants included five patients receiving outpatient haemodialysis and 12 multidisciplinary clinical staff providing care to these patients. Semi-structured interviews were employed and data were analysed thematically. Patients were satisfied with the BIA measurement process and most thought the BIA data would be useful for monitoring changes in their nutrition status. Clinical staff valued BIA data for improving fluid management, assessing nutrition status and supporting patient care. Staff recommended targeting BIA use to patient groups who would benefit the most to improve its uptake in the haemodialysis setting. Conclusions: BIA use in the outpatient haemodialysis setting is acceptable and provides valuable objective data to support health-related behaviour changes in patients and enhance clinical practice. Implementation of BIA should be tailored to the local context and staff should be supported in its use.

## 1. Introduction

End-stage kidney disease (ESKD) is the final stage of chronic kidney disease (CKD), where the kidneys are no longer able to function adequately to clear waste products, electrolytes and excess fluids from the body [1]. Globally, between 4.9 and 9.7 million people are living with ESKD and require a kidney transplant or dialysis to survive [2]. Centre-based haemodialysis (HD) is the predominant therapy, with ≥80% of chronic ESKD patients attending outpatient dialysis clinics for treatment [2].

While dialysis can increase life expectancy by 5–10 years, patients often experience disease and nutrition-related complications such as chronic fluid overload and uraemia, resulting in reduced nutritional intake and involuntary weight loss, and ultimately, malnutrition [1]. Fluctuating body fluid levels coupled with sustained losses of lean muscle mass and fat mass can lead to a significant decline in functional capacity, poor quality of life and increased rates of morbidity and mortality in patients on HD [3]. It is therefore vital to routinely monitor and maintain optimal fluid and nutritional status in HD patients [4,5].

Currently, there is no single ‘gold standard’ method for assessing the hydration and nutrition status of HD patients. Fluid status is normally assessed via clinical observation of intradialytic weight change, pre- and post-dialysis blood pressure, oedema and patient symptoms [6,7]. Nutrition status is assessed by dietitians, using validated tools such as the Patient Generated Subjective Global Assessment (PG-SGA) or Mini Nutritional Assessment (MNA) for elderly patients [8,9]. While these assessment methods are routinely used in clinical practice, each presents limitations. These methods are time-intensive and require the practitioner to have clinical expertise and experience. They are also subjective and prone to intra- and interobserver variability and error. Furthermore, these methods do not objectively assess body composition parameters, which may be useful for identifying and treating subtle changes in fluid and nutritional status in HD patients to reduce the risk of premature morbidity and mortality [10]. Given the significance of ensuring adequate nutrition and appropriate fluid balance in HD patients, incorporating complimentary objective assessment methods into current clinical practice in the HD setting may lead to improved patient care and outcomes. 

Bioelectrical impedance analysis (BIA) can be used to objectively assess body composition, including fat free mass, fat mass, body cell mass and total body water. Advances in technology mean that BIA can be performed with small, portable, multi-frequency devices with scans completed in as little as a few minutes. These devices have been validated for use in clinical populations, including HD patients, and may provide valuable data for clinicians caring for HD patients [11,12]. Indeed, clinical guidelines now recommend the use of multi-frequency BIA (MF-BIA) to assess body composition among adults on maintenance haemodialysis [8]. Despite their potential, little research exists on the acceptability of BIA use in clinical practice, both among HD patients and clinicians caring for them. This includes whether patients would be willing to have regular BIA measures taken (and how often), how useful the data is to both patients and clinicians in self-management/planning care, and whether BIA use is acceptable in an HD setting, where patients and clinicians have a high burden of care. Understanding the perspectives of HD patients and the clinicians caring for them may assist in enhancing care for individuals requiring this life sustaining therapy. 

This study aims to explore the acceptability and clinical value of using BIA to measure the body composition of patients undergoing HD in an outpatient hospital setting from the perspectives of patients and clinicians. 

## 2. Materials and Methods

### 2.1. Study Design

This qualitative descriptive study involved semi-structured interviews that were conducted as part of a larger study investigating the feasibility of using BIA in a HD outpatient setting. Ethical approval was provided by the Gold Coast Hospital and Health Service and Griffith University Human Research Ethics Committees. The study is reported as per the Consolidated Criteria for Reporting Qualitative Research (COREQ) [13].

### 2.2. Setting

The study took place in a day HD unit at a tertiary public hospital in Queensland, Australia. The 20-bed unit provided pre-dialysis and HD services to 75 patients per week. The HD unit was staffed by a multi-disciplinary team which included nephrologists, dialysis nurses and renal dietitians. 

### 2.3. Participants

Participants included patients receiving outpatient HD at the study hospital who were recruited to the main study and the multidisciplinary clinical staff directly involved in providing their care.

Patients were recruited to the larger study by one researcher (KG) with assistance from nursing staff in the unit. Patients were eligible to participate in the main study if they were able to provide informed consent (aged ≥18 years and cognitively intact), could communicate in English (verbally and written) and had been dialysed for ≥three months. Exclusion criteria included concurrent research participation or having a pacemaker in situ. KG consecutively approached eligible patients at their HD appointments and those willing to participate provided written informed consent. As part of the larger study, patients had two BIA measurements (pre- and post-HD) completed during a single dialysis session (reported separately). Pre-dialysis BIA measurements were completed just prior to the commencement of the dialysis session. Post-dialysis BIA measurements were completed approximately 25–30 min following dialysis, with most of this time spent by the dialysis nurses stopping the dialysis session according to standard protocols. Purposive sampling was used to select patients from the larger cohort to ensure variation in age, gender, and years undertaking HD among patients participating in interviews [14]. 

All renal dietitians at the health service were included in the interviews, while purposive sampling was used to select dialysis nurses to ensure that staff of differing roles and years of experience were able to participate in the interviews. Convenience sampling was used to approach nephrologists at the health service about the study via email. Those agreeing to be interviewed provided written informed consent [14].

### 2.4. Data Collection

One author (KG) conducted interviews. She was an honours student at the time, completing a Bachelor of Nutrition and Dietetics. Her previous credentials include a Master of International Public Health with experience in health program management and evaluation, and she was trained in semi-structured interview techniques by an experienced member of the study team (SR). KG had no contact with participants prior to the larger study and was a visiting researcher at the study hospital. Patients were familiar with KG at the time of their interview as she had conducted BIA measurements on all patients during the larger study. 

Patients sat in a comfortable recliner chair during interviews. A semi-structured interview guide was developed based on a framework for assessing the acceptability of healthcare interventions [15] and included three domains: affective attitude; burden; and coherence and perceived effectiveness of BIA (see Appendix A). The interview guide was reviewed and refined by members of the study team prior to interviews. Data were collected directly from patients while undergoing dialysis within two weeks of having their BIA measurements completed.

A separate semi-structured interview guide was developed for staff interviews using the framework described above (see Appendix B). HD nurses had directly observed the BIA measurement process over six weeks of prior data collection, whereas KG described the process to the dietitians and nephrologist prior to their interviews. An example BIA results report was generated from the mBCA seca 525 software utilising de-identified patient data and was provided to all interviewees at the beginning of each interview, which KG explained verbally. All interviews were audio recorded and transcribed for subsequent analysis. Interviews continued until data saturation was reached (i.e., until there were no new emerging ideas or themes) [16]. Repeat interviews were not required and transcripts were not returned to participants for checking due to time constraints.

### 2.5. Data Analysis

Inductive thematic analysis using Braun and Clarke’s six-step guide was employed to analyse interview data [17]. One researcher (KG) transcribed, read and reread transcripts to become immersed in the data, then systematically coded participants’ verbatim statements into short codes. Next, codes were grouped together according to similarity to become themes and subthemes, and these were named and described accordingly. A second member of the study team experienced in qualitative data analysis (SR) checked the groupings and their labels, and using an iterative approach, KG and SR finalised the themes, subthemes and descriptions. For each subtheme, representative participant quotes were selected and are presented alongside the subtheme description. Participants did not provide feedback on the findings due to limited time for data analysis and writing.

Measures put in place to achieve trustworthiness [18,19] included: (a) using a semi-structured interview guide to standardise data collection and an iterative approach to data analysis (credibility); (b) purposively selecting patients of varying age, gender, and years on dialysis, to ensure a range of perspectives were captured (authenticity); and (c) giving detailed descriptions of study setting and context, selection and characteristics of participants, and methods of data collection and analysis (transferability). Finally, reflexivity was considered throughout the study, by the main researcher (KG) considering and reflecting on their own personal bias and potential influence on interactions with participants; and by keeping field notes during and after interviews. 

## 3. Results

### 3.1. Demographics

Five patients participated in interviews; three women and two men with a median age of 70 (IQR 67–77; range 55–85) years. None of the patients approached for an interview declined or were lost to attrition. Two of the patients were on a kidney transplant wait list. Interviews were completed within the first 45–60 min of each patient commencing dialysis to allow the nursing staff time to complete all required tasks and ensure that the patient was able to comfortably settle for the remainder of the dialysis session. Average interview duration was ~12 min (range 6–20 min). Two themes emerged from the patient data: (1) Experience with BIA measurements; and (2) Understanding and the perceived value of BIA measurements (see Table 1).

Twelve multidisciplinary staff directly involved in the care of HD patients also participated, including eight dialysis nurses, three dietitians and one nephrologist. A focus group was held with HD nurses due to the limited time and availability of the nursing staff. Attendees included the Nurse Unit Manager, Clinical Facilitator, two Team Leaders and four dialysis nurses. Two dietitians completed their interview together, due to working in the same unit (job sharing), while the third dietitian and the nephrologist completed their interviews separately. The average staff interview duration was ~20 min (range 14–30 min). Staff responses were represented in three themes: (1) BIA knowledge and skills; (2) BIA use in clinical practice; and (3) Supporting patient education and counselling (see Table 2).

### 3.2. Patient Interviews

#### 3.2.1. Theme 1: Experience with BIA Measurements

Overall, patient responses to having BIA measurements completed pre- and post-dialysis were positive. In this theme, participants described their overall experience with the process, encompassing their perceptions on effort required and burden experienced when completing the measurements, effects on their daily activities, and willingness to repeat the measurements in the future.

(a)Minimal effort and burden: Most patients found it easy to complete both measurements, as P05 said: *“It was extremely easy… I had no problems with it” and it was “no imposition, nothing too difficult or uncomfortable. It didn’t worry me at all”* (P1). Patients said they did not find the process burdensome, particularly because they did not have to make any additional effort to complete the measurement during their dialysis session:


*“The process was well explained, easy enough to do… I didn’t have to do anything, just lay here, simple…it was a pleasure to help. We’re used to being poked and prodded so it’s really not an issue.”*
P2 

One patient commented that the post-dialysis measurement required patients to stay longer than usual: *“Only the fact that I would have to hang around a little bit after dialysis, because I want to get out of here…but it was quick also, so it certainly didn’t put me out”* (P5). However, in general, patients stated that the measurements had little impact on their daily activities because they were scheduled in advance and completed as part of their HD appointment, as P2 indicated: *“no [not burdensome], not at all, it was a pleasure to help. We worked around my dialysis”*.

(b)Engagement and rapport building: Several patients said they enjoyed speaking with the researcher during the BIA measurement process. The minimal effort and burden perceived by patients in having a BIA scan done was complimented by the positive impact of engaging with the researcher, as P1 said: *“I enjoy someone coming and talking to me because (dialysis) is boring… happy to do it again as long as they send you along and we can have a nice chat”*.

#### 3.2.2. Theme 2: Understanding and Perceived Value of BIA Measurements

In this theme, participants’ understanding and perceived value of the BIA data and how it may influence their motivation to change their behaviour were described. 

(c)Understanding of BIA measurements: For most patients, the printed BIA reports were their first exposure to body composition information. Generally, patients had a basic understanding of the results after they were explained by KG: *“I haven’t had a chance to study it yet, but your explanations were good and made me not panic about the red ones [visuals in report]”* (P4); and *“Yeah [I understood], once you explained it to me”* (P2). 

One patient reported some confusion when reviewing the BIA report and suggested simplifying the diagram descriptions generated by the device to reduce confusion and add value for future patients. 

(d)Usefulness to patients and others: Several patients said they found the BIA data interesting; however, they were vague or unsure about how they would use the information for themselves: *“I will find the data useful, I hope”* (P5); and *“It was great to know, nice information”* (P3). Other patients stated that the data would be useful to help ensure they maintained their lean muscle mass and/or to identify other health issues: *“Knowing whether my muscle mass is getting worse. I think it’s good to know that… it’s better to know than not know”* (P4); *“Yes, especially if I started to lose muscle mass. That’s really useful to know”* (P6); and *“I suppose if there was anything in those results that were detrimental to my health, I could use that as a tool to start fixing whatever the issue is”* (P2). All patients indicated that they believed their nephrologist and/or general practitioner would find the BIA data useful. 

Similarly, some patients said they would find it useful to repeat the BIA measurements in order to see if progress had been achieved: “I would have it done again, and then we know how we are progressing” (P3); and “Yeah, of course…it gives me an idea of where I am at right now. Where else would I get that information?” (P4).

(e)Motivation to change behaviour: Patients commonly reported being motivated by the BIA data to engage in behaviours that maintained their lean muscle mass. The reasons for wanting to maintain lean muscle mass varied from a desire to improve general strength to increasing the likelihood of a kidney transplant.


*“I would probably try to get out and walk more or exercise more.” *
P1 


*“It encourages me to think about what else I’m capable of doing…this just puts you in a different mindset. I’ve got a goal; I really need to change to keep my muscle mass on and get as healthy as a I can before this transplant.” *
 P2

### 3.3. Staff Interviews

#### 3.3.1. Theme 1: BIA Knowledge and Skills

This theme describes staffs’ views on the BIA measurement process and resulting data, encompassing their understanding and perceived value of the data to inform clinical decisions and patient care planning. 

(a)Understanding BIA data: Previous exposure to and understanding of body composition parameters varied between nurses, dietitians and nephrologists. While the dietitians and nephrologist were familiar with some or all the BIA measures, nursing staff stated that they had limited understanding of the data and would require additional education to learn how it could be used to inform their care.


*“I guess it’s hard unless you’re actually using it all the time, knowing how to interpret it, and understanding what you are looking at. It would be good to have a session or two… just picking two patients and explaining these reports to the nurses.” *
(P7, nurse) 

(b)Value of BIA data: Dietitians stated that skeletal muscle mass would be the most useful objective measure to monitor in practice and could be used as a comparator to traditional subjective measures to confirm malnutrition: *“I really like the skeletal muscle mass measure… I do think it would be really useful if you are doing it twice a year, to have a look at changes over time and compare it with our PG-SGA assessments”* (D1). Conversely, total body water and extracellular water were considered most useful by the nephrologist, whose primary clinical focus was to remove the patient’s excess body fluid:


*“Assessing a more accurate target weight is the first step, of course. I think knowing exactly how much fluid the patient has got on board would be really valuable.” *
 (P17, nephrologist)

When discussing how the BIA data might be used as part of clinical assessment, nurses and the nephrologist said they would use it to understand a patients’ fluid status rather than focusing on malnutrition: *“Where it could help us is when it’s too dry. That’s the big thing I worry about… it’s hard to know when you’re drying them out. The post-dialysis value would be helpful”* (P6, nurse); *“Mainly in terms of how we achieve euvolemia or not. So often we are so confused about the fluid status, and we can’t just go by what happens on dialysis.”* (P17, nephrologist). Dietitians, on the other hand, stated they would use the data to complement their current subjective assessment methods for identifying and diagnosing malnutrition:


*“We’d like more objective data. Sometimes it’s difficult to distinguish between, for example, if they are losing weight… would they be losing muscle or fat, and it’s a bit hard to…see changes over time and see improvements in muscle mass.” *
(P15, dietitian) 


*“Malnutrition is really important. If someone is dropping their target weights, they’ve been on dialysis, but that target weight continues to drop. Those patients are at more risk of becoming fluid overloaded quickly. We’d like to get on top of it to maximise their nutrition.” *
(P16, dietitian) 

#### 3.3.2. Theme 2: BIA Use in Clinical Practice

In this theme, practitioners discussed their capacity to complete pre- and post-dialysis BIA measures on patients, the perceived barriers and potential solutions to implementing BIA in practice, and potential target patient groups for whom BIA would provide the greatest value.

(c)Barriers and solutions to uptake: While all practitioners expressed an interest in the use of BIA, the majority stated that limited staffing resources, time and competing clinical priorities were the most significant barriers to undertaking BIA assessments in routine practice. Nurses and dietitians indicated that they had limited capacity and that securing additional staffing resources was challenging. For dietitians, having to attend the HD unit at multiple times on the same day to obtain pre- and post-dialysis BIA measurements for patients posed an additional logistical barrier.


*“I think nursing is stretched… there’s a little bit of time in between dialysis and even now that’s quite slim. It’s not to say you couldn’t do that in the future, but if the dietitian was doing them, it’s possible.’ *
(P6, nurse) 


*“I know we do have limited funding, even sometimes just to get the malnutrition audit done, let alone doing these scans—we wouldn’t have time for that.” *
(P15, dietitian) 

Staff also described potential solutions to support BIA uptake in practice. These focused on utilising dietitians or dietetic students in a targeted manner, to perform BIA measurements for a specific purpose; for example, for referrals, assessing high-risk patients, or determining the success of a nutrition intervention.


*“Potentially with us getting dietetic students, it is something that we try and work in every year with our clinical educator and the university.” *
 (P14, dietitian)


*“If we can get a nurse practitioner to do something like this and the dietitian of course. …that’s where we would like dietitians to step in. It’s just that each of us has to play a role, as time is at a premium… that’s where I think we have to work as a team.” *
 (P17, nephrologist)

(d)Target patient groups: All practitioners agreed that the most feasible option for introducing BIA into clinical practice was to target patients who would benefit the most. Four patient groups in the HD setting were identified: high-risk patients (including malnutrition risk), patients with difficult fluid management, patients undertaking or requiring weight loss, and inpatients on the renal ward. 


*“Those difficult fluid patients or the patients we’re concerned about even to start with, our high-risk patients.” *
 (P6, nurse)


*“Maybe there’d be a select group of patients that we use it on… patients that we thought were high risk and then have the BIA done on them even if they weren’t referred.” *
 (P14, dietitian)


*“We’ve got quite a few patients trying to lose weight for kidney transplant, but we want them to maintain their muscle mass, which is a massive thing as well. So that would be really great in that population.” *
(P15, dietitian) 


*“Another area that I’ve been interested in using it, perhaps is in the inpatient setting, on the renal ward. We get patients that are fluid overloaded. Or they’re an inpatient, severely malnourished and we start them on nutritional intervention, and we want to kind of see, are they putting on muscle mass, fat mass, what’s happening to their fluid.” *
(P16, dietitian)

#### 3.3.3. Theme 3: BIA Use to Support Patient Care

This theme reflects staffs’ perceptions of patient receptivity to having BIA measurements completed and the burden that this may place on patients. Staff also described how BIA data could be used in patient education and counselling to improve patient understanding of their condition and motivate them to make positive behavior changes.
(e)Patient receptivity and perceived burden: Nurses generally felt that patients were receptive to having the BIA measurements completed as part of the study because it was a one-off request and some of the most complex patients did not meet the eligibility criteria and were excluded from the study. However, the nurse unit manager (NUM) explained that many of these complex patients were unreceptive to much of the supportive care offered to them, so exclusion from the study may not reflect the willingness of HD patients to undergo BIA measurements.



*“If anything, they loved having the chats with you. If it was going to happen again in another month’s time, they may be like ‘oh no, not that again.” *
(P8, nurse) 


*“I filtered the most difficult ones out for the purposes of the study. But I think with some of those difficult patients, we don’t have much success for supportive care tools. So, we avoid all that extra stuff with them because the wall is up and they’re not going to let the wall down.” *
 (P6, nurse)

(f)Patient understanding and motivation: Nurses and dietitians thought BIA data could be used with the target patient groups described above to improve patient understanding of their medical condition, as well as to reinforce the importance of managing their fluid and nutrition status. The dietitians said a common patient misunderstanding was around the difference between a target dry weight and a healthy body weight. They thought the BIA data could be used to help educate patients on this difference: *“What confuses patients is the discussion around target weight, because the nurses will talk about ‘taking it down’, let’s say from 50 to 48 [kg]… so it’s kind of helping them to understand the difference between a target weight from fluids perspective, versus where the body weight should be from muscle mass and fat mass point.”* (P16, dietitian)

The ability to demonstrate visual progress to a patient via multiple BIA assessments over time was described by nurses and dietitians as the most likely motivator to encourage behaviour change in patients who are either malnourished or trying to lose weight: 


*“With this group of patients, if you show them that one dietary intervention is helping to improve their muscle mass or fat mass or BMI, especially when there is a comparison and it’s pictorial… once you help them to modify their diet and then show them how this has helped them…to reduce the fluid and increase fat or muscle mass, it’s almost like, ‘well done, you’ve done really well’. You can use it as a kind of outcome measure. Nutrition equals survival in dialysis patients.”*
(P16, dietitian)

## 4. Discussion

This qualitative descriptive study explored the acceptability of using BIA for measuring the body composition and fluid status of patients in an outpatient HD setting from the perspectives of patients and clinical staff directly involved in their care. Overall, study findings show that BIA use in this setting was acceptable and valuable to patients and staff. Findings from this study provide insights into the acceptability and clinical value of BIA to HD patients and practitioners and highlight how BIA assessments may be incorporated into routine care in the outpatient HD setting.

HD patients experience considerable levels of treatment burden due to their complex treatment regimens, making it crucial to examine their perspective on any new clinical measure [20,21,22]. The high levels of acceptance towards BIA assessments reported by patients in this study may be related to several factors. First, some of the most complex HD patients were excluded from the eligibility list. These patients were generally unreceptive to additional supportive care tools, as indicated by the nurses. Therefore, the patient acceptability levels reported in this study may represent a ‘best-case’ scenario. Secondly, patients said that they experienced minimal burden with the process, as measurements were taken during a single dialysis session and did not add substantial time to their day. Thirdly, patients expressed high satisfaction with the BIA experience in part due to the opportunity to engage in conversation with the researcher. This was an unexpected finding but is consistent with previous work reporting that hospitalised patients found their interactions with researchers to be one of the most enjoyable aspects of a nutrition study [23]. Finally, patients valued the opportunity to learn about body composition measures and understand the links between their disease and the role of nutrition. Patients indicated that this would motivate them to maintain or improve their body weight and lean muscle mass, consistent with previous research showing that patients enjoyed learning about their nutrition status, and this knowledge motivated them to change their behaviour [23]. This involved providing patients with education to help them better understand their BIA results. In the absence of education, a standalone BIA assessment report is unlikely to result in the same degree of patient acceptability and motivation.

While renal health care professionals generally found the use of BIA in practice to be acceptable, the extent of this varied across different practitioner groups. The level of acceptance was influenced by staffs’ perceptions of their own BIA knowledge and skills, the clinical utility of the data in practice, and potential barriers to uptake. Targeted education was suggested as a first step towards improving practitioners’ understanding of BIA data and how it can be used to support care, particularly for nursing staff. The nephrologist and dialysis nurses valued the opportunity to utilise fluid measures (e.g., total body water) to improve the accuracy of setting patients’ target dry weights and monitor fluid shifts that occur between pre- and post-dialysis. Using BIA primarily to set target dry weights and monitor fluid status during dialysis has been a common focus in previous BIA studies with HD patients [24,25,26]. A systematic review and meta-analysis by Covic et al. examined studies exploring the role of BIA in managing fluid status and establishing dry weight targets in HD patients [24]. Findings from this review indicated that BIA-based assessments for the correction of overhydration had little to no effect on all-cause mortality or changes in body weight [24]. However, BIA assessment methods were positively associated with lower SBP, lower post-dialysis overhydration and reduced arterial stiffness. Therefore, BIA assessment of overhydration may still be perceived as clinically valuable in that it can demonstrate measurable and positive changes in vascular parameters in HD patients [24]. Nevertheless, caution is warranted for utilising BIA solely to guide fluid management in HD patients until further appropriately designed, randomised controlled trials are carried out [27] Dietitians in the present study viewed skeletal muscle mass as a useful parameter to support better nutrition screening and assessment practices, aligning with current recommendations in the National Kidney Foundation’s Clinical Practice Guideline for Nutrition in CKD [8] and the GLIM criteria [28], which recommend using BIA in the assessment and diagnosis of malnutrition in adult HD patients and acute clinical populations, respectively. This, alongside the interest in and acceptance of BIA by practitioners in this study, creates a case to support the adoption of BIA in the outpatient HD setting. However, with the introduction of any new intervention, perceived barriers to uptake must be considered and its use should be practical, efficient and supported to improve patient outcomes. 

Renal care professionals have previously reported limited self-efficacy due to lack of BIA knowledge and skills, and reduced capacity to manage the dialysis care process, as barriers to systematic uptake and use of BIA in practice [29]. In the current study, nurses and dietitians identified limited staffing and clinical time, and competing clinical priorities as the most significant barriers to uptake. Consequently, two solutions were recommended: using a multidisciplinary approach to BIA assessment that includes dietitians, dietetic students, dialysis nurses, and nephrologists; and targeting BIA in high-risk patients, difficult fluid patients, patients undertaking or requiring weight loss, and inpatients. Each renal care professional plays a critical role in providing care to HD patients to ensure optimal outcomes are achieved. Previous research has shown that multidisciplinary care models for patients with CKD can lower all-cause mortality and reduce hospitalisations [30]. By adopting a multidisciplinary approach and strategically targeting its use to the four patient sub-groups identified, BIA can be integrated successfully and sustainably into standard clinical practice in the outpatient HD setting [12]. 

### 4.1. Limitations

It is important to acknowledge that this was a single-site study with 17 participants, so the findings may not be generalisable to other clinical settings. Additionally, the interviewer in this study was the same person who completed the BIA measurements in the larger feasibility study; therefore, patients and staff may have been less likely to express negative perceptions to the interviewer. Acknowledging this, the researcher encouraged participants to be as honest as possible and explained the importance of honest feedback for the study results. Furthermore, due to the time pressures on the HD nursing staff, a focus group discussion was held instead of individual interviews. Differences in perceived status between the participants, for example, between the Nurse Unit Manager and the dialysis nurses, may have influenced some responses. To ensure that participants engaged in the discussion and felt comfortable expressing their opinion, the researcher emphasised the importance of hearing all viewpoints from the beginning, remained neutral, acknowledged more dominant participants’ opinions, sought out further information from quieter participants and kept the discussion focused. Also, the Nurse Unit Manager noted that, during the study period, the dialysis nurses had the most exposure to the BIA device and encouraged their input during the group discussion. Lastly, due to the limited timeframe for this study, not all dietitians, nurses and nephrologists involved in the care of the patients could be interviewed. Consequently, not all patient and practitioner views may have been represented in the data. To combat this, participants were purposively selected to include patients of differing ages, genders and years on dialysis; and staff of differing roles and years of experience, in order to ensure that a range of perspectives were captured.

### 4.2. Implications and Recommendations

Findings from this study provide valuable insights into the perceived barriers and facilitators for using BIA in the outpatient HD setting, potentially informing the implementation of BIA use in routine care in adult HD patients. For any hospital-based setting contemplating using BIA in clinical practice, it is important to consider providing training to ensure that clinical staff are equipped with the knowledge and skills required to complete and interpret BIA measurements and explain the results to patients. It is also recommended that target patient groups are identified and prioritised for BIA measurements, and that a multidisciplinary care approach is adopted to optimise the use of resources in any setting. 

Future research should include implementation studies to evaluate the uptake and utilisation of BIA by clinicians in HD settings. Involving the target end-users (clinicians) early in the implementation of new interventions allows for iterative improvements to be made in real-time, and may enhance acceptability and expedite the adoption of BIA in clinical practice [31].

## 5. Conclusions

This study evaluated the acceptability and perceived barriers and enablers for using BIA to measure body composition and fluid status in adult HD patients from the perspectives of patients and staff. Results suggest that BIA is acceptable to these groups and provides valuable data to support health-related behaviour changes and enhance patient care.

## Figures and Tables

**Table 1 healthcare-10-01205-t001:** Patient interview findings.

Themes	Subthemes
1. Experience with BIA measurements	(a) Minimal effort and burden (b) Engagement and rapport building
2. Understanding and perceived value of BIA measurements	(c) Understanding of BIA measurements (d) Usefulness to patients and others (e) Motivation to change behaviour

**Table 2 healthcare-10-01205-t002:** Staff interview findings.

Themes	Subthemes
1. BIA knowledge and skills	(a) Understanding BIA data (b) Value of BIA data
2. BIA use in clinical practice	(c) Barriers and solutions to uptake (d) Target patient groups
3. BIA use to support patient care	(e) Patient receptivity and perceived burden (f) Patient understanding and motivation

## Data Availability

The study data is available upon request from the corresponding author.

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
