# Peer review of "Patient and Staff Perceptions on Using Bioelectrical Impedance Analysis in an Outpatient Haemodialysis Setting: A Qualitative Descriptive Study"

_healthcare, 2022, doi:10.3390/healthcare10071205_

Round 1

Reviewer 1 Report

This article is a qualitative study exploring patient and staff acceptability and perceived value of BIA in an outpatient haemodialysis setting. I recognize that limited research exists on the acceptability and utility of BIA in clinical practice. Thank you for the opportunity to review this manuscript, which is very well-written with a nice flow, interesting and relevant information, and great applicability to practice.

I do however have a few suggestions. On page 1 rows 33-36 I believe the citation needs to be amended. Please see suggestion below:

Globally, between 4.9 and 9.7 million people are living with ESKD and require a kidney transplant or dialysis to survive [2]. Centre-based haemodialysis (HD) is the predominant therapy, with ≥80% of chronic dialysis ESKD patients attending outpatient dialysis clinics for treatment [2].

You adhere very nicely to the COREQ checklist. However, would it not be relevant to refer to Tong (2007) and the checklist already in the study design section on page 2 row 84? Could you also please add a reference for your method - qualitative research - when mentioning “purposive sampling” page 3 row 102, “convenience sampling” page 3 row 107, data saturation page 3 row 132, “Measures put in place to achieve trustworthiness” page 4 row 147, etc?

I assume the interviews with the patients were performed during dialysis. However, I think this should be more clearly described. Since people treated with HD often report intra dialytic fatigue please provide information about when (if during dialysis) the interviews were conducted. Was it in the beginning or near the end of the treatment? Furthermore, as hierarchical structures and different levels of education may hinder participants from expressing their opinions in focus groups, as for example described by Tausch and Menold (2016) doi: 10.1177/2333393616630466. Considering the mix of participants in your focus group (manager, team leaders and dialysis nurses), I think this matter should be discussed.

You repeatedly refer to the “limited time” (page 4 row 145, page 4 row 166, page 9 row 423). Why the rush, and did this circumstance affect the quality of the study a lot, or why do you put such emphasis on the fact?

On page 3 row 101 and on page 5 row 192, you describe that BIA measurements had been performed (pre- and) post HD, and this is presented as a potential barrier. I would find it relevant to learn for how long the patients were asked to stay after dialysis for the post HD measurement, as there is no consensus in the literature about when post HD measurements should be performed. Furthermore, on page 7 row 285 you mention the pre and post measurements again as a presumable logistic barrier. Could it be discussed that sometimes only pre measurements are performed and accepted, for feasibility reasons?

On page 3 row 130 and page 5 row 208 you mention “an example BIA report”. What report is this, is it generated from the Fresenius Fluid Management Tool of was is something your team put together? On page 6 row 252 and page 9 row 392 you report on the perceived value of BIA data. It makes me curious on what data the device provided. Could you find a place to mentioning which device was used?

Finally, I believe it would be of clinical relevance to discuss the results reported on page 6 row 259 “I think knowing exactly how much fluid the patient has got on board would be really valuable”, for example as discussed in an editorial by David (2016) doi: 10.2215/CJN.01770220, in which he talks about using BIS for fluid management being a complex intervention.

Author Response

The authors would like to thank Reviewer 1 for taking the time to review our paper and provide constructive feedback. Our responses to your comments are provided in the attached document.

Reviewer 2 Report

Even so the authors nicely presented their questionnaire study concerning BIA some important corrections should be done.

1.       The authors should point out in more detail the pitfalls of a BIA.

2.       Point 2.1. XX Hospitals XX University? Please fill in which one

3.       What does as part of a lager study mean, which larger study, please explain in more detail

4.       In 2.3. Data Collection -> the paragraph about the student does not relate to this research paper and should therefore be deleted.

Author Response

The authors would like to thank Reviewer 2 for taking the time to review our paper and provide feedback. Our responses to your comments are provided in the attached document.

Reviewer 3 Report

This manuscript is well-written and interesting.

One minor concern is that I would recommend the authors to include references to support the methods (" Study design"  "Data collection" and "data analysis" ).

Author Response

The authors would like to thank Reviewer 3 for taking the time to review our paper and provide feedback. Our responses to your comments are provided in the attached document.

Round 2

Reviewer 2 Report

The authors provided the additional Information I asked them fore.